# Footprint of Domestic Processing on Safety and Functional Properties of Italian Black Garlic

**DOI:** 10.3390/foods14152595

**Published:** 2025-07-24

**Authors:** Davide Addazii, Chiara Cevoli, Flavia Casciano, Federico Ferioli, Tullia Gallina Toschi, Andrea Gianotti, Lorenzo Nissen

**Affiliations:** 1DiSTAL–Department of Agricultural and Food Sciences, Alma Mater Studiorum—University of Bologna, Food Science Campus, P.za G. Goidanich 60, 47521 Cesena, Italy; davide.addazii2@unibo.it (D.A.); chiara.cevoli3@unibo.it (C.C.); cascianoflavia@yahoo.it (F.C.); federico.ferioli@unibo.it (F.F.); tullia.gallinatoschi@unibo.it (T.G.T.); 2CIRI-Interdepartmental Centre of Agri-Food Industrial Research, Alma Mater Studiorum—University of Bologna, P.za G. Goidanich 60, 47521 Cesena, Italy

**Keywords:** *Allium sativum*, maturation process, gastronomy, mechanical strength, volatile compounds, prebiotic

## Abstract

Garlic (*Allium sativum* L.) is extensively recognized for its health-promoting effects and functional attributes, including antibacterial and anti-inflammatory activities. Additionally, the derived product of the industrial maturation process, known as black garlic, is famous for its functional properties. The novelty of the present work is to characterize the functional properties of domestically produced black garlic. In fact, this study examines the domestic maturation of fresh garlic bulbs into black garlic of two Italian varieties, focusing on microbial growth, antimicrobial properties, prebiotic activity, volatile organic compounds, mechanical resistance, brown intensity, pH, and Aw. Results show that domestic processes are microbiologically and chemically safe and generate black garlic products with functional attributes such as prebiotic activity and the presence of health-related bioactive compounds, also developing superior technological performance. These findings enhance the understanding of black garlic culinary practices, leveraging gastronomic preparations for the development of healthier and safer food products.

## 1. Introduction

Garlic (*Allium sativum* L.) is a member of the *Alliaceae* family and is a worldwide staple food with several health-related attributes [1,2]. Food and applied scientists have highlighted the functionalities of garlic, referring, for example, to antibacterial, anticancer, anti-inflammatory, antioxidant, and blood lipid-regulating activities, also considering the heritage practices of traditional medicine that led to the production of effective pharmaceutical products [3]. Garlic is rich in sulphur compounds, flavonoids, and bioactive compounds that contribute to its antibacterial, anti-inflammatory, cardiovascular protection, and immunostimulant activities [4]. Among gastronomic preparations derived from heritage culinary practices, the maturation of fresh garlic into black garlic is an easily accessible domestic approach that is largely diffused in Asian cuisines but seldom reproduced in Europe. Although this maturation process modifies fresh food organoleptic characteristics, reducing and transforming pungent sulphur compounds, such as allicin, to odorless and milder sweet-sour flavors, e.g., DAS, it enhances the nutritional properties and eventual bioactivity [1]. Black garlic, whose origin is debated to be from Japanese, Korean, and Chinese sources, is produced by subjecting fresh garlic to a maturation process that can last up to 21 days at temperatures between 60 and 90 °C and humidity levels ranging from 60 to 80% [5,6]. The water-soluble antioxidant S-allyl-L-cysteine, which is the allicin compound in black garlic forms, has been shown to have anti-inflammatory, anticarcinogenic, neuroprotective, anti-allergic, and antidiabetic properties [1,7,8]. 5-hydroxymethylfurfural (5-HMF), an intermediate of the Maillard reaction, is also produced during the process. It has been connected to a number of health issues, such as carcinogenicity, tumor formation, and liver and kidney toxicity. Its LD50 value has been reported to be 871.12 mg kg^−1^ [6]. Interestingly, HMF exhibits antioxidant activity, inhibits immune-related allergic responses, suppresses xanthine oxidase activity, and contributes to the prevention of sickle cell hemoglobin formation [9]. From a commercial perspective, the garlic market is massive and in a rising trend, and concurrently the black garlic market is also experiencing important market expansion, and the black garlic market size is forecast to reach $200.0 million by 2030, at a CAGR of 7.50% during the forecast period 2024–2030 [10]. This trend is largely fueled by growing consumer interest in gourmet experiences and health-oriented food choices. In Western Europe, in particular, black garlic occupies a premium niche, with strong demand from fine dining establishments and consumers seeking functional food products. Although precise market figures are not available, current trends suggest steadily increasing interest and a positive market forecast for black garlic across the region. So far, it is estimated that also domestic production of black garlic will be propagated to Western kitchens. Although, in literature, there is a lack of knowledge on domestic preparation of black garlic in terms of food safety, composition, and quality. There is poor evidence on how the organic compounds are transformed and which final compounds are obtained. Also, there is no track of functional attributes, such as prebiotic potential, antimicrobial activity, and retainment of bioactive compounds.

This research aims to contribute to a more comprehensive understanding of the transformations that are triggered throughout the black garlic production process, with a focus on evaluating the impact of processing technologies and the role of microbial safety in the maturation of black garlic. The objective is to explore whether the transformation of white garlic into black garlic influences some of its functional properties (e.g., prebiotic activity). This work also considered the quantification of the main microbial groups that characterized the indigenous flora of garlic to highlight the microbial safety of the final products. To this end, two varieties of garlic, Sulmona (SG) and Voghiera (VG) of DOP labels, were analyzed comparing their effects on food-borne pathogenic or spoilage microorganisms. This study, for the first time, compares the black garlic process of Sulmona and Voghiera varieties and provides new knowledge that may be useful for optimizing the production of black garlic and enhancing its value as a functional ingredient in gastronomical preparations. In fact, while black garlic has been widely studied for its functional properties and industrial production methods, little attention has been given to domestic-scale processes. This study addresses that gap by exploring non-industrial black garlic production, offering new insights into how household procedures can influence the product’s functional qualities. By focusing on artisanal methods, this research contributes to expanding current knowledge and provides valuable information for culinary professionals, food technologists, and health-conscious consumers interested in sustainable and accessible food innovations.

## 2. Materials and Methods

### 2.1. Garlic Samples and Maturation Process

Two varieties of garlic (*Allium sativum* L.), Sulmona garlic (AS) and Voghiera garlic (AB), were used for the study. The transformation process to black garlic was conducted using a multipurpose fermenter (Black Garlic Fermenter Machine 90 W, Standard Hardwa, Taiyuan, China). Each fermenter was loaded with 1 kg of garlic bulbs, and the process parameters were set to maintain a temperature of 75 °C and a relative humidity of 80% for a period of 12 days. In order to observe the change in various parameters such as pH, Aw, color, moisture, and texture, samples were taken at different times during the process. Samples (*n* = 3) were taken before treatment, after six days, and at the end of the process. Results were obtained from 3 biological replicates performed for each variety of garlic.

### 2.2. Microbial Sampling

Microbiological analysis was conducted using the plate count technique to quantify viable microbial cells, serving as indicators of microbiological food safety. The methodology followed the standards ISO 4833-1:2013 [11], ISO 21528-2:2017 [12], and ISO 17410:2019 [13] for the enumeration of mesophilic bacteria, yeasts and molds, and psychrotrophic bacteria, respectively, with minor modifications. At each sampling point, 10 g of garlic were collected from three biological replicates. Two technical replicates were prepared by independently homogenizing the samples with 90 mL of sterile saline solution (0.9 g/L NaCl) using sterile filtered bags (Interscience, Saint Nom La Bretèche, France) and processed in a stomacher apparatus (Seward, Worthing, UK) at 120 paddle/min for 2 min. Serial dilutions of the homogenates were prepared in sterile saline solution using glass tubes. Selected dilutions were plated on sterile agar-based media (Oxoid, Thermo Fisher, Waltham, MA, USA) and incubated under specific conditions: mesophilic bacteria were enumerated on Plate Count Agar (PCA) at 30 °C for 48 h, while yeast and molds were enumerated on SDA (Sabouraud Dextrose Agar) enriched with chloramphenicol at 0.1 g/L (Merck, Darmstadt, Germany) at 30 °C for 24–48 h and psychrophiles on PCA at 5 °C for 7–10 days. The microbial load was calculated as the mean of three biological replicates (*n* = 3) and expressed as Log CFU/g (Colony Forming Units per gram) ± standard deviation.

### 2.3. Microbial Strains and Growth Conditions

The following list of bacteria was used to evaluate different aspects of the functionality of garlic samples, i.e., the in vitro assessment of antimicrobial activity, minimum inhibitory concentration (MIC) and prebiotic potential. The strains used in the determination of MIC were probiotic microbial strains obtained from commercial cultures (Bromatech, Albese, CO, Italy) or deposited type strains or probiotic-like strains belonging to the Culture Collection of DISTAL at the University of Bologna. The list included (i) *Listeria innocua* as a surrogate for *Listeria monocytogenes* [14] (Department of Agricultural and Food Sciences of the University of Bologna), *Bacillus amyloliquefaciens* DSM 1060, *Escherichia coli* ATCC 25922, and *Escherichia coli* NCIMB 555 as food-borne and spoilage microorganisms; (ii) a probiotic lactobacilli mix containing *Lactobacillus acidophilus* LA14, *Lactobacillus acidophilus* LA1, *Lacticaseibacillus rhamnosus* C1112, *Lacticaseibacillus rhamnosus HN001*, and *Limosilactobacillus reuterii* LR92; and (iii) a mix of probiotic bifidobacteria *Bifidobacterium bifidum* NCIMB 700795, *Bifidobacterium lactis* BL-04, *Bifidobacterium bifidum* BB-06, *Bifidobacterium breve* BB-03, *Bifidobacterium longum* BL-05; (iv) a mix of aerobic probiotics *Enterococcus faecium* UBEF-41 and *Lactiplantibacillus plantarum* subsp. *plantarum* NCIMB 8299. The microorganisms used for prebiotic assessment were all probiotic microbial strains obtained from commercial cultures (Bromatech, Albese, CO, Italy) or deposited type strains or probiotic-like strains belonging to the Culture Collection of DISTAL at the University of Bologna. *E. coli* strains were deposited as type strains. Probiotics have been previously isolated from commercial supplements and repeatedly propagated in our laboratory [15]. *Lactiplantibacillus plantarum*
*NCIMB 8299*, *Lactobacillus acidophilus* LA1, *Lacticaseibacillus rhamnosus* C1112, *Lacticaseibacillus rhamnosus* HN001, *Limosilactobacillus reuterii* LR92, *Bifidobacterium bifidum* NCIMB 700795, *Bifidobacterium lactis* BL-04, *Bifidobacterium bifidum* BB-06, *Bifidobacterium breve* BB-03, *Bifidobacterium longum* BL-05, *E. coli* ATCC 25922, and *E. coli* NCIMB 555 were cultured from glycerol stocks stored at −80 °C and were propagated in selective media (Oxoid, Thermo Fisher Scientific, Waltham, MA, USA). The strains used in the determination of the MIC were prepared using an inoculum at Log 6 CFU/mL in a, brought to level with the corresponding growth substrate, the concentration was calculated by absorbance value at OD_600_, with UV-VIS LLG-uniSPEC 2 spectrophotometer (Mettler-toledo Spa, Milano, Italy), using an internal standard curve. The selective broths and agars (Oxoid, Thermo Fisher Scientific, Waltham, MA, USA) used to propagate the strains were: MRS (De Man, Rogosa and Sharpe) for aerobic probiotic mix, MRS + 0.05 g/L cysteine for probiotic lactobacilli and probiotic bifidobacteria mix, TSB (tryptone soy broth) for *Bacillus amyloliquefaciens* and *Lis. innocua*, and BHI (Brain Heart Infusion Broth) for *E. coli* strains. For prebiotic potential, cultivation and propagation of bacteria was made according to previous reports [15]. Lactobacilli mix was counted on MRS (Man-Rugosa-Sharpe) agar after propagation for at least 24 h at 37 °C in jars with anaerobiosis catalyst (Oxoid, USA). Bifidobacteria mix was counted on MRS agar supplemented with 0.05 g/dL L-cysteine (Merck, Germany) after propagation in the same conditions of lactobacilli. *E. coli* mix was counted on BHI (brain-heart infusion) agar (Oxoid, USA) after propagation at 37 °C for 24 h. Strains were all maintained at −80 °C in 20% (*v*/*v*) glycerol (Merck) with the appropriate medium. From refrigerated stocks, bacteria were propagated for two consecutive passages, before preparing the working suspensions at Log 7 CFU/mL.

### 2.4. Evaluation of Minimum Inhibitory Concentration (MIC) Versus Food Borne Pathogens and Intestinal Beneficial Bacteria

MIC was determined by adopting the broth dilution method as previously described [16]. Briefly, garlic-to-water suspensions (*w*/*v*) were prepared as 20% (*w*/*v*) solutions in distilled and sterile water and used promptly afterwards. Serial 1:2 geometric dilutions, ranging from 10.0% to 0.625% (*v*/*v*), were performed directly in a 96-well round-bottom microtiter plate (Greiner, BioOne. Srl, Milano, Italy) using an appropriate broth; controls were included in each plate. Wells containing 200 µL solutions were inoculated with a microbial suspension at the concentration of Log 6 CFU/mL. The samples used were garlic samples prior (AB and AS) and after the maturation process (Abt12 and Ast12). Garlic samples were prepared by dividing each bulb into cloves, adding sterile distilled water, and homogenizing through a stomacher apparatus for 2 min to obtain solutions at a concentration of 20% (*w*:*v*) [16]. Chloramphenicol and Rifaximin (Merck, Germany) in a 20% (*w*:*v*) solution were used as controls at the maximum concentration of 256 ug/uL. *Lis. innocua*, *E. coli* 555, and *Bac. amyloliquefaciens* were incubated in aerobiosis for 24 h at 37 °C, while the lactobacilli mix and the bifidobacteria mix were grown at 37 °C for 24 h in jars with anaerobic catalyst (Thermo Fisher Scientific, USA). MIC results were evaluated with a spectrophotometer multiplate reader (mod. Spark, Tecan Ltd., Mannedorf, Switzerland) after 24 h of incubation.

### 2.5. Evaluation of Prebiotic Score of Garlic Samples

Garlic samples were used for calculating the prebiotic score with the related formula from two biologically independent experiments and technical triplicates as previously described [15], including qPCR quantifications. 1 mL of suspension of finely sieved garlic samples in sterile distilled water (1:10 *w*:*v*) was placed in 9 mL of broth in order to obtain concentrations at 1 g/dL. 1 g/dL of FOS (frcuctooligosaccharides) and 1 g/dL of Inulin (INU) from chicory (Merck, Germany) were used as a prebiotic positive control, and 1 g/dL of glucose was used as a negative control. The microorganisms used were contained in three bacterial mixes: (i) a probiotic lactobacili mix, (ii) a probiotic bifidobacteria mix, and (iii) a mix of two pathogenic food-borne *E. coli*. All bacterial mixtures were used at the final concentration of Log 6 CFU/mL.

### 2.6. Quantification of Bacterial DNA by qPCR

Microbial DNA ofmedia and the prebiotic activity assay was extracted with the Spin Food Nucleus DNA Extraction Kit (Macherey-Nagel, Darmstadt, Germany). Genetic standards for qPCR were prepared from serially diluted (1:10 *v*:*v*) PCR amplicons using the ProFlex PCR system (Applied Biosystems, Foster City, CA, USA) and SuperFi Platinum Taq (Invitrogen, Carlsbad, CA, USA) and purified with the GeneJet PCR purification kit (Thermo Fisher Scientific, USA). Absolute quantitative PCR was performed with QuantStudio 5 (Applied Biosystems, USA) and QuantStudio Design and Analyze 2.1 software (Applied Biosystems, USA). Reactions were applied as previously described [15] and are reported as Appendix A.

### 2.7. pH, Aw, and Moisture Content

pH analysis was carried out in triplicate with a pH meter (Mettler Toledo, Italy), calibrated with standard buffer solutions of pH 4.00, 7.00, and 10.00. Water activity was measured with the AquaLab apparatus (Decagon Devices Inc., Pullman, WA, USA), as previously described [17]. Moisture content was determined by gravimetry to constant weight with AOAC Official Method 925.10.

### 2.8. Brown Intensity and Mechanical Properties

Garlic color was measured by using a CIELab colorimeter (Konica Minolta Meter CR-400, Tokyo, Japan). The browning intensity (ΔE) was calculated as ΔE = [(ΔL) 2 + (Δa) 2 + (Δb) 2]1/2, where ΔL = L − L*, Δa = a − a*, and Δb = b − b*. L, a, and b represent the colors of the samples; and L*, a*, and b* represent the colors of the base at time zero [18,19]. Texture determination was conducted with a TA-Hdi^®^ Texture Analyzer instrument (Stable Microsystems, Godalming, UK) with an installed 25 kg load cell equipped with a 5 mm diameter probe, set at a descent speed of 0.5 mm/sec. From the force-deformation curve, obtained from the instrument, maximum force values, expressed in Newtons (N), were extrapolated.

### 2.9. Volatilome Analyses by Solid-Phase Microextraction–Gas Chromatography–Mass Spectrometry (SPME-GC-MS)

The volatilome of garlic samples was characterized through the analysis of volatile organic compound (VOC) profiles, with the aim of identifying bioactive volatile molecules potentially involved in prebiotic activity. VOC determination was performed using an Intuvo Agilent 7890 A gas chromatograph coupled to an Agilent 5975 mass spectrometer (Agilent Technologies, Santa Clara, CA, USA), operating in electron impact mode at an ionization voltage of 70 eV. The protocols for SPME-GC-MS analysis and compound identification were applied as previously described [15]. Prior to sampling, the SPME fiber was conditioned by exposure to the GC inlet for 10 min at 250 °C in a blank sample for thermal desorption. 4-methyl-2-pentanol (Merck, Germany) was employed as an internal standard at a final concentration of 20 mg/kg and was left equilibrating at 40 °C for 10 min in a water bath. The SPME fiber was exposed to each sample for 40 min and then inserted into the injection port of the GC for 10 min for sample desorption. The temperature program was: 50 °C for 1 min, then programmed at 1.5 °C/min to 65 °C, and finally at 3.5 °C/min to 220 °C, which was maintained for 25 min. Injector, interface, and ion source temperatures were 250 °C, 250 °C, and 230 °C, respectively. Injections were carried out in spitless mode and helium (3 mL/min) was used as carrier gas. Identification was obtained with the NIST 11 MSMS library and the NIST MS Search program 2.0 (NIST, Gaithersburg, MD, USA).

### 2.10. Data Processing and Statistical Analysis

All statistical analyses were conducted using Statistica 8.0 software (Tibco Inc., Palo Alto, CA, USA). The assumption of normality was assessed using the Shapiro–Wilk test, while homoscedasticity was evaluated through Levene’s test [20]. Differences among sample groups were examined via multivariate analysis of variance (MANOVA), and relationships among variables were further explored using principal component analysis (PCA), followed by post hoc Tukey’s honestly significant difference (HSD) test and Duncan’s post hoc for prebiotic activity (*p* < 0.05) [21]. Organization of the dataset, normalization, and preparation of PCA were conducted as previously presented [22]. Pairwise comparison and Venn diagrams were elaborated with the digital tool available at https://molbiotools.com/listcompare.php (last accessed on 16 July 2025). All results are expressed as mean values ± standard deviation obtained at least from duplicate batches in three independent experiments.

## 3. Results and Discussion

### 3.1. Evolution of the Growth of Endophytic Mesophiles and Yeasts

Endophytic microbial abundance was assessed by plate count, sampling prior to, over, and at the end of the maturation process. AS and AB had similar levels of both eubacteria and yeast and molds, mainly derived from the external peels of garlic bulbs. It is interesting to note that the maturation process and the high temperatures produced were able to generate an effective reduction in the viability of yeast and molds, already before day 6, faster than the reduction of the viability of eubacteria, which occurred after day 6 (Table 1). In scientific literature it is reported that the process is considered microbiologically safe, reducing the presence of any microbes to their quantitation limits. Generally, an inhibition is already seen after the very first days of the process, and a drastic reduction in viability is seen after a week. The final product at the end of the process seldom hosts microbes, whose eventual quantitation is limited to levels of less than Log 3 CFU/g [23].

### 3.2. pH, A_w_, and Moisture

Table 1 gives an overview of the pH, water activity (Aw_w_), and moisture content values recorded at different times during the production process of the two garlic varieties. During the different process steps, a clear decreasing trend is shown for pH, Aw and moisture content. In particular, pH showed a significant decrease over time. In fresh samples (t0), values were relatively similar between the two varieties; after six days of maturation (t6), there was a marked decline in pH that became even more evident after twelve days (t12). The decrease of the pH observed during the process was generally attributed to the formation of organic acids, that were absent in the fresh sample and were generated during the maturation process. In fact, at the end of the maturation process, black garlic turned rich in acetic, formic, succinic, and 3-hydroxypropionic acids that, in line with other reports, reduced pH [24]. Aw also followed a similar dynamic, with a gradual decrease over the course of ripening. Fresh samples showed almost identical Aw for both varieties. At day 6, Aw decreased moderately and then declined more markedly at day 12. Previous studies have reported higher Aw values than those observed in this particular process. For example, Ref. [25] Katarzyna et al. (2020) emphasized the need to further reduce Aw to ensure microbiological stability of the product. They suggest that Aw values below 0.85 can be achieved by further drying the garlic after the fermentation process, providing a less favorable environment for unwanted microbial growth and ensuring greater food safety in the final product. Finally, moisture content also showed an important decrease over time, declining significantly as early as day 6 until it reached its lowest values at day 12. It should be noted that not all the studies in literature have reported similar moisture contents, also because the duration of the process in these studies was longer than the 12 days [18,26,27]. Moisture is an important process parameter in black garlic production, which influences the sensory characteristics, nutritional value, and biological activity of the product [26]. During heat treatment, the moisture content decreases due to water evaporation [19]. Moisture content in the range of 45–54% promotes the development of a favorable texture structure of black garlic (flexible and chewy texture) [19]. Other authors showed that moisture in garlic maturation has a significant effect on evaporation rate, retained moisture content, and appearance (70–90% *w*/*v*) and texture of black garlic [28].

### 3.3. Brown Intensity and Mechanical Properties

No significant differences were found between the two products in terms of color change over the observation period (Figure 1a). The Maillard reaction is responsible for the remarkable color change from the white color of raw garlic to the dark brown color characteristic of black garlic [29]. This phenomenon is the result of the formation of several compounds, including 5-HMF, melanoidin, and other brown polymeric compounds generated during heat treatment [30]. The color change index (ΔE) increased rapidly with growing temperature because the reactivity of the Maillard reaction is closely related to the temperature of the heat treatment [31]. To monitor the dynamics of color change during the 12 days of treatment, ΔE was calculated as a quantitative index according to the following formula:
${\Delta E=[(\Delta L)2+(\Delta a)2+(\Delta b)2]}^{1\backslash 2}$. Regarding the results of texture (Figure 1b) shown in fresh garlic, there was a greater applied force than on day 6, where a decrease was observed. This variation can be attributed to the maturation process of the garlic, which involves an increase in its softness in the early stages, followed by a progressive development of the chewy texture towards the end of the cycle. In fact, an increase in the force applied was observed on day 12. During the process, the texture of the product changed, becoming more rubbery, soft, and delicate. As a result of these processes, a completely different product was created, which gained in sensory attractiveness [25].

### 3.4. Antimicrobial Properties

MIC results are shown in % ranging from 10% to 0.625% (*w*:*v*) of broth dilutions (Table 2). The most relevant MIC values were observed in the fresh garlic samples (ASt0 and ABt0), where moderate antimicrobial activity emerged against several strains; in particular, ASt0 had a slightly higher antimicrobial activity than ABt0. In contrast, samples at the end of the process (ASt12 and ABt12) showed scarce antimicrobial activity. Compared with the antibiotics used as controls, white garlic samples showed visible, though less effective, antimicrobial activity, while black garlic samples exerted almost no inhibition to the extent of the in vitro test used, showing a decline in antimicrobial efficacy after the maturation process. The antimicrobial activity of garlic is mainly due to the action of allicin [32,33], which easily permeates the cell membrane and reacts with the thiol groups inside the cell, leading to death [33]. This mechanism is inhibited in mammalian eukaryotic cells by the presence of intracellular glutathione. Studies have shown that allicin has the ability to inhibit several strains and species, such as *Escherichia coli*, *Pseudomonas* spp., *Staphylococcus* spp., *Micrococcus* spp., *Salmonella* spp., *Listeria monocytogenes,* and *Bacillus* spp. [34]. The reason why black garlic has limited antimicrobial activity is because allicin is thermolabile and is degraded during the production of the black garlic sample. Eventually, for future improvement of the process lower temperature could be applied. The little antimicrobial activity of black garlic is therefore due to other compounds. Studies show that hydroxymethylfurfural (HMF) has antimicrobial activity not only because of its ability to inhibit certain enzymes involved in glycolysis, including glyceraldehyde-3-phosphate dehydrogenase and alcohol dehydrogenase, but also because of its denaturing effect on microbial plasmids [35,36]. Melanoidins are also produced by the Maillard reaction, which exhibits chelating activity with respect to Fe3^+^ and Mg2^+,^ two metals with direct implications in the metabolism and membrane structure of various microbial species [37].

### 3.5. Prebiotic Activity

Firstly, considering the results of the bacterial quantification trend (Table 3), the growth of *E. coli* in ABt0 and ASt0 samples was lower than that observed in ABt12 and ASt12 (Table 3). Among this feature, there is a clear difference between fresh and black garlic samples for both varieties. This reduction may be attributed to the antimicrobial activity of allicin and sulphur compounds, which primarily act against Gram-negative bacteria [33,38]. Regarding bifidobacteria, significant differences were observed during the process between samples ABt0 and ABt12, with a variation of nearly 1 Log. As for lactobacilli, a slight but statistically significant difference in growth was noted between fresh and black garlic samples. Specifically, for both garlic varieties, higher bacterial growth values were recorded in samples ABt12 and ASt12. Thus far, we have found that (i) fresh garlic was able to reduce *E. coli* mix, while black garlic was not, in particular for ABt12; (ii) fresh garlic fostered more than black garlic the growth of bifidobacteria mix, in particular for ABt12; and (iii) black garlic fostered more than fresh garlic the growth of lactobacilli mix for both garlic varieties. The higher attraction to black garlic samples of lactobacilli and partly of bifidobacteria in comparison to that for fresh garlic samples could be explained by the content and quality of sugars prior to and after the maturation process. In fact, according to [29], the free sugar composition of black garlic compared to fresh garlic increases 16-fold [29]. In particular, in black garlic the predominant sugars are fructose and glucose, while in fresh garlic the predominant sugar is sucrose. During the transformation to black garlic, melanoidins also occur, and it is reported that these molecules can contribute to prebiotic activity [37,38,39].

The index of prebiotic activity of positive control (FOS) against the mixture of lactobacilli and bifidobacteria was overcome by the white garlic samples (Table 4). Black garlic obtained lower results than the fresh samples and the controls for both probiotic mixtures, with values much lower than those of FOS and inulin. The variety with the lowest values corresponded to ASt12. The index obtained for ASt0 remained the same for both bifidobacteria and lactobacilli mixes, placing it among the samples with higher prebiotic activity. On the other hand, the index of ABt0 was higher for lactobacilli than for bifidobacteria. Similar results were observed by other authors that reported the stimulation effect of melanoidins toward the growth of beneficial bacteria [40]. Also, it was reported that black garlic showed to enhance the growth of *Bifidobacterium* spp. in the gut, with effects that appear to be enterotype-specific, suggesting that black garlic may modulate the gut microbiota and probiotic populations differently depending on the individual’s baseline microbial composition, potentially offering targeted benefits for gut health [41].

### 3.6. Volatile Organic Compounds Related to the Transformation Process

Volatile garlic compounds are generated from odorless and non-volatile alk(en)yl cysteine sulfoxides (ACSOs) that accumulate during storage. After tissue disruption by crushing, cutting, or chewing, ACSOs under the action of the alliinase enzyme give rise to unstable thiosulphinates that rapidly decompose and generate aromatic components [42]. To thoroughly investigate the differences at the level of volatile molecules during the process, the volatilome was characterized by SPME-GC-MS. Venn diagrams show a clear evolution of volatile composition over time. In the case of AB samples (Figure 2a), a progressive redistribution of compounds was observed, with 16 volatile metabolites shared among all ripening stages, while some compounds were exclusive to each sampling time (24 unique for ABt0 and ABt6). A similar trend was also observed for AS samples (Figure 2c), although with greater variability between stages, which was evidenced by the reduced overlap between different ripening times. Analysis with Jaccard’s index (Figure 2b,d) quantified these differences. For AB garlic, the Jaccard value between ABt6 and ABt12 was 0.418, indicating moderate similarity, while for AS garlic, the value between ASt6 and ASt12 reached 0.475, suggesting a more pronounced evolution of volatile composition over time. Comparative analysis between the two varieties at different ripening times (Figure 2e,f) shows convergence in volatile composition as ripening time increases. In particular, although at t0 the difference between the two cultivars was more pronounced, with little overlap in compounds, after twelve days (t12) the Jaccard index between ABt12 and ASt12 was 0.654, the highest value recorded in the analysis, suggesting that ripening contributed to the uniformity in the volatile profile of the two varieties.

Analysis of the garlic volatilome revealed significant transformations in the composition of volatile organic compounds (VOCs) during the ripening process. Through the use of principal component analysis (PCA), it was possible to observe changes in the chemical profiles of the two garlic varieties studied. The first two analyses (Figure 3a,b) focused on bioactive volatile compounds, such as organic acids, aldehydes, alcohols, and ketones. The PCA results show that fresh garlic samples (t0) exhibited a distinctive profile compared to those subjected to ripening for six (t6) and 12 days (t12). Over the duration of the process, the compounds redistributed differently between the two varieties, resulting in a distinct separation in the volatile profiles. ANOVA loadings, on the other hand, highlighted which compounds had a greater impact on the differentiation of the samples. Among these, Orcinol, Nicotinyl alcohol, Phenol, and 2-Furancarboxaldehyde were found to be particularly abundant in the mature samples, suggesting the role of these compounds in the aromatic evolution of garlic over time. In fact, other authors recently described the presence of these molecules in black garlic and the importance of preserving them using encapsulation technology [43].

Also, from ANOVA loading of just processed garlic samples, unique signature for ASt12 were assigned to Nicotinyl alcohol and 2-Furancarboxaldehyde, while unique signature for ABt12 were assigned to Orcinol, Pyrazine, 2,6-dimethyl, and Pyrazine methyl. This trend is in line with what has been observed by [44], who identified an enrichment of aromatic compounds such as furfural and its derivatives in black garlic resulting from the Maillard reaction. Special attention was then paid to sulphur compounds, which are responsible for garlic’s characteristic aroma and biological properties [42] (Figure 3c,d). PCA shows a clear separation between fresh and ripened samples, signaling a profound reorganization of the sulpho-compound profile as ripening progresses. Again, the two varieties showed a distinct maturation trajectory, indicating varietal differences in the metabolism of these compounds. ANOVA loading analysis confirmed the key role of certain sulpho-compounds in characterizing mature volatilome, including diallyl sulphide (DAS), dimethyl disuphide (DADS), and dimethyl trisulphide (DATS), along with compounds such as 3-vinyl-1,2-dithiacyclohexen-5-one, which could contribute to the formation of flavors. Similarly, Ref. [44] reported a reduction in primary sulpho-compounds in black garlic, with a significant decrease in DADS and diallyl trisulphide DATS in the early stages of ripening, while compounds such as allyl methyl sulphide showed a relative increase. Also, from ANOVA loading of just processed garlic samples, a unique signature for ASt12 was assigned to 2,4-Dithiapentane, while unique signatures to ABt12 were assigned to 1,2-Dithiolane and 1,3,5-Trithiane. In a recent investigation, Ref. [42] focused on the determination of the aromatic profile of different types of Italian garlic and elephant garlic (*Allium ampeloprasum* L.), including the two varieties employed in the present research, and results are in line with our findings. In fact, results from SPME analyses confirmed disulphides as the main chemical class among garlic volatiles, representing on average more than 80% of these compounds. Diallyl disulphide was the major individual component, ranging from 32.7 to 67.6% of volatiles. Vinyldithiins and trisulphides were also detected in significant amounts after SPME extraction, amounting on average to 8.5 and 6.7% of aromatic compounds, respectively. With regard to AS, allyl methyl disulphide, diallyl disulphide, allyl (E)-propenyl disulphide, 2-vinyl-4H-1,3-dithiin, and diallyl trisulphide were the most abundant sulphides determined in that study, accounting for 4.4, 43.9, 28.9, 7.8, and 5.1% of total volatiles, respectively, whereas the same compounds were at levels of 3.2, 67.6, 18.0, 3.5, and 3.4%, respectively, in AB. Significant correlations were highlighted between the amount of the main volatiles determined by SPME and their non-volatile precursors, namely methilin, alliin, and isoalliin. Our results, relative to these sulpho-compounds, are also comparable to other recent reports; one focused on nine clones of fresh garlic and black garlic derivative [45], another that uses black garlic as an ingredient to enhance functional and sensorial attributes of bread [46], and a review reporting black garlic production in Turkey [47].

## 4. Conclusions

This study provides a comprehensive assessment of the domestic maturation process of black garlic, focusing on microbial growth, chemical transformations, antimicrobial properties, prebiotic activity, volatile organic compounds (VOCs), and some technological features, which is a novel aspect of the existing knowledge of this process and product because it highlights domestic procedures. Thus, this research adds new knowledge about the non-industrial black garlic production and the product’s functional properties that could be beneficial to culinary experts, food technologists, and the general public. The results indicate that while black garlic retains some prebiotic functionality after fermentation (especially in ABt12, which had a mean of 0.20 and 0.33 for the lactogenic and the bifidogenic activities, respectively), its activity is lower than that of fresh garlic. This reduction is likely related to sugar transformations during ripening, including an increase in fructose and glucose, and the formation of melanoidins through the Maillard reaction. Although melanoidins are known for their prebiotic effects, their contribution in this context appears to be less significant compared to fresh garlic. In terms of antimicrobial properties, black garlic showed very little antimicrobial activity to the extent of the in vitro test, mainly due to allicin degradation. Nevertheless, other bioactive compounds produced during fermentation, such as sulphur derivatives (DAS, DADS, DATS), 5-HMF, and melanoidins, still offered moderate antimicrobial effects through mechanisms such as enzyme inhibition and metal ion chelation. VOCs and organic acids produced during the process also likely play a role in the overall bioactivity of the product, although further research is needed to explore their influence on the gut microbiota. From a technological perspective, fermentation resulted in notable changes in texture and color, making the product softer and darker, which are improvements that contribute to its sensory appeal and market value. In conclusion, black garlic is confirmed to be a promising functional food with retained, although reduced, prebiotic and other functional benefits. Moreover, this is the first time it has been reported that a domestic process can yield a product comparable to that obtained through industrial-scale methods in approximately half the time. Future studies should focus on in vitro gut microbiota interactions to understand the impact on intestinal ecology and integrative analyses of fermentation metabolites to better understand their biological relevance. Also, a critical review should be conducted on consumer acceptability and sensory implications. These approaches may pave the way for expanding the domestic use of black garlic and promoting healthier, more informed food choices.

## Figures and Tables

**Figure 1 foods-14-02595-f001:**
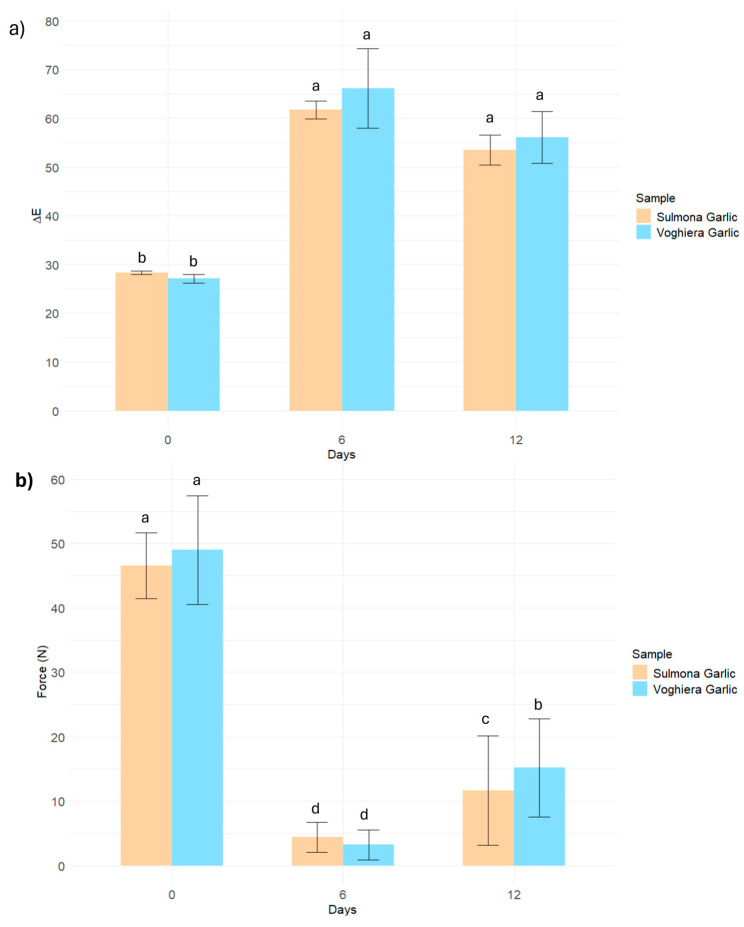
(**a**) Measurement during the process of brown intensity and (**b**) mechanical properties. Letters over the barplots indicate statistical singificant differences within a graph, by ANOVA and *post hoc* Tuckey’s HSD test with *p* < 0.05.

**Figure 2 foods-14-02595-f002:**
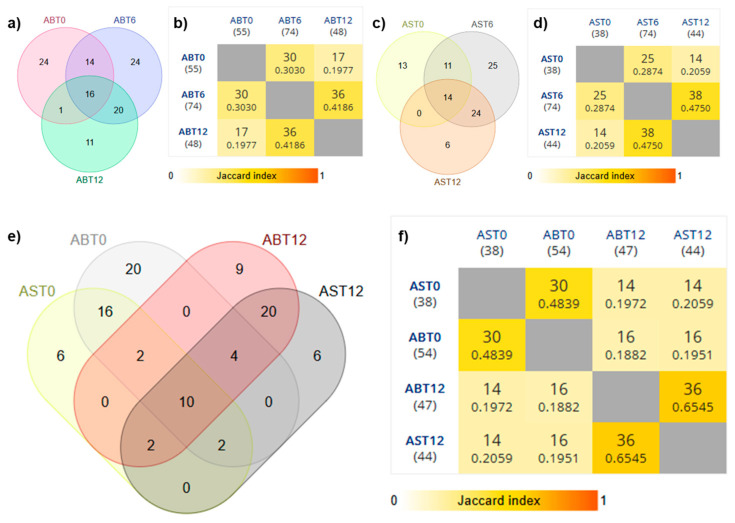
Volatilome compositions of garlic samples by the theory of sets. Comparison among samples and among the time points of the maturation process. (**a**,**c**,**e**) Venn diagrams; (**b**,**d**,**f**) pairwise comparison tables; AB = Voghiera garlic sample; AS = Sulmona garlic sample; t0 = garlic sample not processed; t6 = six days of maturation process; t12 = 12 days of maturation process; Supporting statistical tables can be found as Appendix A.

**Figure 3 foods-14-02595-f003:**
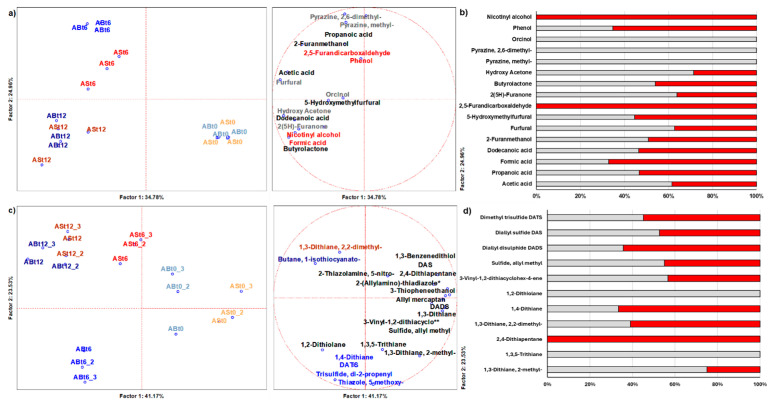
Volatile organic compounds (VOCs) of the volatilome of garlic samples during the maturation process. (**a**) Principal component analysis (PCA) of bioactive VOCs of garlic samples during the maturation process and (**b**) multivariate analysis of variance (MANOVA) loadings of bioactive VOCs of garlic samples at the end of the maturation process; (**c**) PCA of sulphurated VOCs of garlic samples during the maturation process and (**d**) MANOVA loadings of sulphurated VOCs of garlic samples at the end of the maturation process. AB = Voghiera garlic sample; AS = Sulmona garlic sample; t0 = garlic sample not processed; t6 = six days of maturation process; t12 = 12 days of maturation process; DAS = diallyl sulphide; DADS = disulphide, dimethyl; DATS = dimethyl trisulphide; * 2-(Allylamino)-thiadiazole = 2-(Allylamino)-5-ethyl-1,3,4-thiadiazole; ** 3-Vinyl-1,2-dithiacyclo = 3-Vinyl-1,2-dithiacyclohexen-5-one. Supporting statistical tables can be found as Appendix A.

**Table 1 foods-14-02595-t001:** pH, Aw, moisture and microbial load of garlic samples during the maturation process.

Analysis	Garlic Samples at Different Times of Maturation						
	ABt0	ASt0	ABt6	ASt6	ABt12	ASt12	*p*-Value
pH	5.89 ± 0.09 ^a^	5.97 ± 0.05 ^a^	4.48 ± 0.07 ^b^	4.64 ± 0.07 ^b^	3.68 ± 0.05 ^c^	3.76 ± 0.07 ^c^	0.000001
Aw	0.979 ± 0.002 ^a^	0.978 ± 0.010 ^a^	0.855 ± 0.072 ^a,b^	0.872 ± 0.021 ^b^	0.63 ± 0.013 ^c^	0.712 ± 0.011 ^c^	0.000001
Moisture (%)	62.61 ± 5.10 ^a^	65.07 ± 1.71 ^a^	40 ± 14.21 ^b^	36.19 ± 7.80 ^b,c^	28.09 ± 9.11 ^c^	22.25 ± 9.81 ^c^	0.000001
Yeast *	5.84 ± 0.33	4.30 ± 0.75	<10.00 ± 0.00	<10.00 ± 0.00	<10.00 ± 0.00	<10.000 ± 0.00	0.476000
Mesophiles *	6.55 ± 0.07 ^a^	6.44 ± 0.02 ^a^	6.99 ± 0.15 ^b^	6.98 ± 0.25 ^b^	<10.00 ± 0.00 ^b^	<10.00 ± 0.00 ^b^	0.000002

* Log CFU/g. AB = Voghiera garlic sample; AS = Sulmona garlic sample; t0 = garlic sample not processed; t6 = six days of maturation process; t12 = 12 days of maturation process. Different letters at values apex indicate statistical significance within a row by ANOVA followed by Tukey’s post hoc test with *p* < 0.05.

**Table 2 foods-14-02595-t002:** Minimal inhibitory Minimal inhibitoryconcentration of garlic samples diluted in selective bacterial broths % (v/v).

Strains	Minimal Inhibitory Concentration % (vol:vol)						
	ASt12	ABt12	ASt0	ABt0	Rifaximin	Chloramphenicol	*p*-Value
*Listeria innocua*	>10.00 ± 0.00 ^a^	>10.00 ± 0.00 ^a^	2.50 ± 0.00 ^c^	5.00 ± 1.44 ^b^	0.62 ± 0.00 ^d^	0.62 ± 0.00 ^d^	0.000066
*Escherichia coli*	>10.00 ± 0.00 ^a^	>10.00 ± 0.00 ^a^	1.25 ± 0.72 ^b^	1.25 ± 0.00 ^b^	0.62 ± 0.00 ^c^	0.62 ± 0.00 ^c^	0.000233
*Bacillus amyloliquefaciens*	>10.00 ± 0.00 ^a^	>10.00 ± 0.00 ^a^	2.50 ± 0.00 ^c^	5.00 ± 0.00 ^b^	0.62 ± 0.00 ^d^	0.62 ± 0.00 ^d^	0.00451
Aerobic probiotic mix	>10.00 ± 0.00 ^a^	>10.00 ± 0.00 ^a^	1.25 ± 0.72 ^b,c^	2.50 ± 0.00 ^b^	<0.62 ± 0.00 ^c^	0.62 ± 0.00 ^c^	0.000741
Probiotic bifidobacteria mix	>10.00 ± 0.00 ^a^	>10.00 ± 0.00 ^a^	5.00 ± 1.44 ^b^	10.00 ± 0.00 ^a^	0.62 ± 0.00 ^c^	0.62 ± 0.00 ^c^	0.000006
Probiotic lactobacilli mix	>10.00 ± 0.00 ^a^	>10.00 ± 0.00 ^a^	2.50 ± 0.00 ^c^	5.00 ± 1.44 ^b^	0.62 ± 0.00 ^d^	0.62 ± 0.00 ^d^	0.000515

AB = Voghiera garlic sample; AS = Sulmona garlic sample; t0 = garlic sample not processed; t6 = six days of maturation process; t12 = 12 days of maturation process. Different letters at values apex indicate statistical significance within a row by ANOVA followed by Tukey’s post hoc test with *p* < 0.05.

**Table 3 foods-14-02595-t003:** Bacterial enumerations on selective broths and 1% *w*/*v* of garlic suspensions by plate-count and qPCR methods.

Garlic Samples	Mean ± SD (Log Cells/mL)		
	*Escherichia coli* Mix	Bifidobacteria Mix	Lactobacilli Mix
ASt12	10.31 ± 8.9 ^c^	10.93 ± 9.23 ^c^	11.36 ± 10.12 ^c^
ABt12	12.77 ± 11.12 ^a^	10.34 ± 9.22 ^d^	11.08 ± 9.53 ^d^
ASt0	7.95 ± 6.13 ^d^	10.88 ± 9.12 ^c^	10.91 ± 9.04 ^d^
ABt0	7.97 ± 6.24 ^d^	11.28 ± 9.81 ^b^	10.70 ± 8.93 ^e^
Glucose	10.94 ± 9.01 ^b^	11.35 ± 8.82 ^b^	11.63 ± 10.25 ^b^
FOS	10.27 ± 8.82 ^c^	12.32 ± 11.01 ^a^	12.30 ± 11.22 ^a^
*p* value	0.000344	0.004664	0.000252

Glucose = negative control; FOS (Fructo-oligosaccharides) = positive control. Garlic preparations were resuspended in sterile MilliQ^®^ water. AB = Voghiera garlic sample; AS = Sulmona garlic sample; t0 = garlic sample not processed; t6 = six days of maturation process; t12 = 12 days of maturation process. Different letters at the values apex indicate statistical significance within a column by ANOVA followed by Duncan’s post hoc test with *p* < 0.05.

**Table 4 foods-14-02595-t004:** Prebiotic index toward probiotic mixes.

Garlic Samples	Mean ± SD Prebiotic Index	
	Lactogenic Effect	Bifidogenic Effect
ASt12	0.08 ± 0.04 ^d^	0.13 ± 0.03 ^d^
ABt12	0.33 ± 0.05 ^c^	0.20 ± 0.04 ^d^
ASt0	0.67 ± 0.05 ^a^	0.67 ± 0.05 ^a^
ABt0	0.74 ± 0.03 ^a^	0.63 ± 0.02 ^a^
FOS	0.51 ± 0.03 ^b^	0.53 ± 0.04 ^b^
INU	0.37 ± 0.01 ^c^	0.39 ± 0.04 ^c^
*p* value	0.000231	0.000066

FOS (Fructo-oligosaccharides) = positive control. INU (Inulin) = positive control. Garlic samples were resuspended in sterile MilliQ^®^ water. AB = Voghiera garlic sample; AS = Sulmona garlic sample; t0 = garlic sample not processed; t12 = 12 days of maturation process. Different letters at the values apex indicate statistical significance within a column by ANOVA followed by Duncan’s post hoc test with a *p*-value < 0.05.

## Data Availability

The original contributions presented in the study are included in the article/Appendix A, further inquiries can be directed to the corresponding authors.

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
