# Peer review of "Footprint of Domestic Processing on Safety and Functional Properties of Italian Black Garlic"

_foods, 2025, doi:10.3390/foods14152595_

Round 1
Reviewer 1 Report
Comments and Suggestions for Authors
The co-authors have examined the domestic maturation of fresh garlic bulbs into black garlic of two Italian varieties, focusing on microbial growth, antimicrobial properties, prebiotic activity, volatile organic compounds and technological features.
The article needs major revision as the current form is not acceptable for publication.
I have following comments and suggestions:
Abstract: The abstract needs improvement. The presented results should be consistent with the results section.
Main body
- Please emphasize the novelty of the results and their significance for science and practice.
- Compare your results deeply with international studies to clarify the context in the discussion portion.
- Provide more up-to-date references presenting similar research results.
- The resolution of the pictures must be increased to improve the visibility of the markings on the pictures. Especially figure no 3.
- In my opinion, the weakest part of the publication is the discussion.Please compare your research results with those of other authors, emphasizing differences, similarities and the complexity of the research. The methodology section is much longer and more detailed than the discussion of results.
- It is good practice to describe results and use the past tense rather than the present tense.
- The work has great scientific potential. However, it requires reinforcement in the discussion section.
Author Response
The co-authors have examined the domestic maturation of fresh garlic bulbs into black garlic of two Italian varieties, focusing on microbial growth, antimicrobial properties, prebiotic activity, volatile organic compounds and technological features.
The article needs major revision as the current form is not acceptable for publication.
I have following comments and suggestions:
Abstract: The abstract needs improvement. The presented results should be consistent with the
results section.
We thank the referee for the revision and we have now provided a revised abstract
Main body
1. Please emphasize the novelty of the results and their significance for science and practice.
We thank the referee for the comment and we have now added more phrases relative to the novelty and significance of the work, both in introduction and in conclusion sections.
2. Compare your results deeply with international studies to clarify the context in the discussion portion.
We thank the referee for the note. In the revised version we have discussed deeper the results obtained with other literature 2024-2025 references, adding 6 more references
3. Provide more up-to-date references presenting similar research results.
We have added 6 new references of recent years 2024-2025 presenting similar results in the sections 3.5 and 3.6
4. The resolution of the pictures must be increased to improve the visibility of the markings on the pictures. Especially figure no 3.
Figure 1 has been replaced with an harmonized one. Figure 3 is provided in high resolution and does not blurry at 400% magnification on the word document.
5. In my opinion, the weakest part of the publication is the discussion. Please compare your research results with those of other authors, emphasizing differences, similarities and the complexity of the research. The methodology section is much longer and more detailed than the discussion of results.
We thank the referee for the comment, and we have added and discussed more similar results.
6. It is good practice to describe results and use the past tense rather than the present tense.
We apologize for the inconvenience, and we have amended the verb tense.
7. The work has great scientific potential. However, it requires reinforcement in the discussion section.
We thank the referee for the compliment and we have tried to reinforce the discussion, also with 6 new references comparisons.

Reviewer 2 Report
Comments and Suggestions for Authors
Dear Authors,
The article and the research that use produced add new knowledge about the black garlic production and functional properties that could be beneficial to the, as you also pointed out, to culinary experts, food technologist but also public. Although it is slightly different aim of the study I would suggest authors read the paper: https://doi.org/10.1002/fci2.70002 and https://doi.org/10.1002/mbo3.547.
There are minor issues I would advise to address.
- Abstract
Line 20, what does the “technological features” mean.
Line 20 Instead of “procedures” I would advise production processes.
Line 22 word fine bioactive properties are misused, rephrase it.
- Suggest editing Key words to more appropriately address the topic.
- Introduction
For the introduction, in the line 34 word plenty is wrongly used to give information on the richness of the compounds in the garlic.
Line 42 origin is debated to be
Line 44 I would suggest giving precise number of days for the maturation process.
Line 55 it is estimated…
Aim of the study lack defining microbial safety aspects of your research, later there is great emphasis on it in Methodology and Discussion.
I would suggest addition of identified microorganisms in garlic and black garlic, also opinion on possible safety issues.
- Methods
Methods are appropriate, clearly described and could be repeated. Methods are standard for this research.
In the section 2.10 I would suggest adding the explanation of Jaccard index used later for volatile components analysis and adding mean ± SD or standard deviation, that is later used as a data representation in the tables.
- Results and Discussion
In all the tables I would suggest expressing mean ± SD and add p value where missing (only given in Table 1).
Overall Tables and Figures are clear and appropriately used.
Lines 361-364 sentence should be rephrased to follow English meaning and sentence logic.
- Conclusions
Conclusions follow the aim and the results of the research.
Line 488 This section should be omitted.
Kind regards
Comments on the Quality of English LanguageDear Authors,
I would suggest reviewing and editing english in the article by a professional, there were some major errors in wording and sentence developing.
Author Response
The article and the research that use produced add new knowledge about the black garlic production and functional properties that could be beneficial to the, as you also pointed out, to culinary experts, food technologist but also public. Although it is slightly different aim of the study I would suggest authors read the paper: https://doi.org/10.1002/fci2.70002 and https://doi.org/10.1002/mbo3.547.
We thank the referee, and we have also included some discussion compared to the suggested references.
There are minor issues I would advise to address.
1. Abstract
Line 20, what does the “technological features” mean.
We have revised that specifying the analyses conducted, as mechanical resistance, brown intensity, pH and Aw
Line 20 Instead of “procedures” I would advise production processes.
Corrected
Line 22 word fine bioactive properties are misused, rephrase it.
Replaced with “health-related bioactive compounds”
2. Suggest editing Key words to more appropriately address the topic.
We have replaced some keywords: maturation process, gastronomy, mechanical strength; volatile compounds; prebiotic
3. Introduction
For the introduction, in the line 34 word plenty is wrongly used to give information on the richness of the compounds in the garlic.
We have replaced “plenty” with “rich”
Line 42 origin is debated to be
Corrected
Line 44 I would suggest giving precise number of days for the maturation process.
Corrected with last up to “21 days”
Line 55 it is estimated…
Corrected
Aim of the study lack defining microbial safety aspects of your research, later there is great emphasis on it in Methodology and Discussion.
We have now included the term “microbial saferty”
I would suggest addition of identified microorganisms in garlic and black garlic, also opinion on possible safety issues.
This work also considered the quantification of the main microbial groups that characterized the indigenous flora of garlic to highlight also microbial safety of the final products. This final aspect has been reported more intensively in the revised version.
4. Methods
Methods are appropriate, clearly described and could be repeated. Methods are standard for this research.
Thanks, indeed!
In the section 2.10 I would suggest adding the explanation of Jaccard index used later for volatile components analysis and adding mean ± SD or standard deviation, that is later used as a data representation in the tables.
We thank the reviewer and we have added this information. “Pairwise comparison and Venn diagrams were elaborated with the digital tool available at https://molbiotools.com/listcompare.php (last accessed on 07/16/2025). All results are expressed as mean values ± standard deviation obtained at least from duplicate batches in three independent experiments.”
5. Results and Discussion
In all the tables I would suggest expressing mean ± SD and add p value where missing (only given in Table 1).
We thank the reviewer and we have added the required information
Overall Tables and Figures are clear and appropriately used.
Lines 361-364 sentence should be rephrased to follow English meaning and sentence logic.
We have rephrased and made clearer the section as: This reduction may be attributed to the antimicrobial activity of allicin and sulphur compounds, which primarily act against Gram-negative bacteria [30,35]. Regarding bifidobacteria, significant differences were observed during the process between samples ABt0 and ABt12, with a variation of nearly 1 Log. As for lactobacilli, a slight but statistically significant difference in growth was noted between fresh and black garlic samples. Specifically, for both garlic varieties, higher bacterial growth values were recorded in samples ABt12 and ASt12.
6. Conclusions
Conclusions follow the aim and the results of the research.
Line 488 This section should be omitted.
Corrected
Comments on the Quality of English Language
Dear Authors,
I would suggest reviewing and editing english in the article by a professional, there were some major errors in wording and sentence developing.
We have checked and corrected the grammar, making the text more accurate and fluent.

Reviewer 3 Report
Comments and Suggestions for Authors
Overall evaluation
The methodology used is correct, but I do not share the conclusions regarding an antimicrobial or prebiotic activity in black garlic.
Major remarks
Liner 491-492. Contrary to what the authors said, table 4 does not show any probiotic effect of black garlic, but only a limited activity of white garlic.
Line 506. There is not a retained anti-microbial activity, as can be seen in Table 2, the MIC for black garlic is > 10., i.e. very low.
In Table 3, contrary to what is reported in the title, the microbial growths obtained with the plate sowing technique but not with the PCR-count are reported.
In Table 3 it is not clear whether the statistical comparison is performed separately for each microbial strain or considering them all together. In my opinion, it is better if the statistical analysis is performed separately for each bacterium.
Minor remarks
Line 79: In my opinion the word “with” at the end of the sentence “ and a relevant humidity of 80% for a period of 12 days., with.” should be removed.
Line 488. Please erase the words “This section” at the beginning of the paragraph.
Author Response
Referee #3
Overall evaluation
The methodology used is correct, but I do not share the conclusions regarding an antimicrobial or prebiotic activity in black garlic.
We thank the referee and we have revised discussion with antimicrobial or prebiotic activity of black garlic. For example, the antimicrobial activity of black garlic is now reported in discussion as not effective to the extent of the in vitro test used. Otherwise, prebiotic activity is reduced in comparison to fresh garlic, but still appreciable in ABt12 as it is higher than 0.00.
Major remarks
Liner 491-492. Contrary to what the authors said, table 4 does not show any probiotic effect of black garlic, but only a limited activity of white garlic.
We are sorry, but we disagree, since prebiotic activity is higher than 0.00 also for ASt12 and ABt12, thus we have reported this sentence in brackets to be clearer “(especially in ABt12, that had a mean of 0.20 and 0.33 for the lactogenic and the bifidogenic activities, respectively)”
Line 506. There is not a retained anti-microbial activity, as can be seen in Table 2, the MIC for black garlic is > 10., i.e. very low.
We thanks the referee and we have included this corrected sentence: ”black garlic showed very little anti-microbial reduced activity to the extent of the in vitro test”.
In Table 3, contrary to what is reported in the title, the microbial growths obtained with the plate sowing technique but not with the PCR-count are reported.
The values reported in table 3 are means of Pasteurian and Molecular microbiology, as previously reported.
In Table 3 it is not clear whether the statistical comparison is performed separately for each microbial strain or considering them all together. In my opinion, it is better if the statistical analysis is performed separately for each bacterium.
We thank the referee and we have corrected Table 3 and the foot notes
Minor remarks
Line 79: In my opinion the word “with” at the end of the sentence “ and a relevant humidity of 80% for a period of 12 days., with.” should be removed.
Corrected
Line 488. Please erase the words “This section” at the beginning of the paragraph.
Corrected

Reviewer 4 Report
Comments and Suggestions for Authors
Minor revision:
- Consider simplifying the title for clarity: “Impact of Domestic Processing on Safety and Functional Properties of Italian Black Garlic”
- Strengthen the rationale for domestic processing by citing consumer trends or gaps in European adoption.
- Clarify the novelty: Is this the first comparative study of Sulmona vs. Voghiera garlic in black garlic transformation?
- Specify the model and supplier of the fermenter used.
- Clarify whether qPCR quantification was absolute or relative.
- While qPCR offers insight into bacterial growth, further interpretation on microbiota modulation (especially in vivo relevance) would strengthen the functional food claims.
- The volatile and prebiotic profiles of Sulmona vs. Voghiera garlic could be highlighted more distinctly to deepen varietal comparisons.
- Discuss why black garlic loses antimicrobial activity, could this be mitigated by process optimization?
- The Venn diagrams and PCA are excellent, but consider summarizing key compound shifts in a simplified table for readers less familiar with VOCs.
- The conclusion is well-rounded but could benefit from a brief mention of future applications (e.g., culinary innovation, nutraceuticals).
- Suggest including a sentence on consumer acceptability or sensory implications.

Author Response
Referee #4
Minor revision:
- Consider simplifying the title for clarity: “Impact of Domestic Processing on Safety and Functional Properties of Italian Black Garlic”
We thank the referee and we have followed the suggestion with a little modification, reporting the new title as “Footprint of Domestic Processing on Safety and Functional Properties of Italian Black Garlic”.
- Strengthen the rationale for domestic processing by citing consumer trends or gaps in European adoption.
New sections at lines 65-71 and 91-97 have been added in the revised MS
- Clarify the novelty: Is this the first comparative study of Sulmona vs. Voghiera garlic in black garlic transformation?
This novelty aspect has been reported at line 88 in the revised MS
- Specify the model and supplier of the fermenter used.
Added
- Clarify whether qPCR quantification was absolute or relative.
Clarified at line 223
- While qPCR offers insight into bacterial growth, further interpretation on microbiota modulation (especially in vivo relevance) would strengthen the functional food claims.
We thank for this suggestion and we have written something about it as future perspectives in conclusion.
- The volatile and prebiotic profiles of Sulmona vs. Voghiera garlic could be highlighted more distinctly to deepen varietal comparisons.
- Discuss why black garlic loses antimicrobial activity, could this be mitigated by process optimization?
Added at line 401
- The Venn diagrams and PCA are excellent, but consider summarizing key compound shifts in a simplified table for readers less familiar with VOCs.
We thank the referee for the note, in fact we have originally provided supplementary tables reporting the names of molecules for each set or subset.
- The conclusion is well-rounded but could benefit from a brief mention of future applications (e.g., culinary innovation, nutraceuticals).
We thank for this comment and we have reported this feature in the revised version
- Suggest including a sentence on consumer acceptability or sensory implications.
Added at line 616
